# Europeanisation in the Field of Housing: Its Areas of Influence, Different Approaches, Mechanisms, and Missing Links

Jorge Afonso *[ID] and Paulo Conceição [ID]

CITTA—Research Centre for Territory, Transports and Environment, Department of Civil Engineering, Faculty of Engineering, University of Porto, Rua Dr. Roberto Frias s/n, 4200-465 Porto, Portugal; psc@fe.up.pt
* Correspondence: up201105640@up.pt

**Abstract:** Housing has been, and continues to be, a central concern of economic, geographical and political research, as well as of social debates. It is worth noting that the European Union (EU) does not possess exclusive or shared competence in the field of housing. Rather, its influence is the result of policies in other areas. Building on the call in the literature to examine both Europeanisation and housing studies, we present and discuss the areas of the EU's influence—economic, environmental, legal, political, social, and urban. The literature shows that these influences have resulted from different shifts in the European agenda, with different approaches (top-down, bottom-up) and mechanisms (legislative, economic and fiscal, cognitive), creating conflicting housing narratives. In conclusion, future research should focus on understanding the influences of member states as well as the intersection between housing and other policy areas. Additionally, the distribution and transfer of power in decision-making within the EU should be examined, as well as the strategic interactions between (housing) political actors from different member states and EU institutions, and the consequences of such interactions.

**Keywords:** housing; European Union; Europeanisation





## 1. Introduction

Housing has been, and continues to be, a central concern of economic, geographical and political research, as well as of social debates (Kleinman et al. 1998; Clapham et al. 2012). The aesthetic, cultural, and symbolic value of housing is recognized, as is its complexity, its multidimensional nature, its fixity and durability, and its diversity across place, time, and social norms.

Nevertheless, in the field of housing, the European Union does not possess exclusive or shared competence (Allegra et al. 2020; Krapp et al. 2020; Kucharska-Stasiak et al. 2022). This may be attributed, at least in part, to the hybrid understanding of housing as both a market commodity and a public good (Bengtsson 2001; Tosics and Tulumello 2021). Each Member State has its own unique organization of housing provision (Krapp et al. 2020) under the principles of subsidiarity and proportionality. Both principles define the relationship between the EU and its member states. The first principle states that the EU should only undertake necessary tasks or those it can perform most effectively. The second principle is that the EU should not go beyond what is necessary to achieve its objectives (Best 2016; McCormick 2022). Furthermore, assuming exclusive or shared competence in the field of housing can be expensive for EU resources (Kucharska-Stasiak et al. 2022).

This is not a novel phenomenon. The hybrid understanding of housing and the lack of exclusive or shared competence are not novel either. This suggests that the European Union's impact on housing provision is a consequence of its policies in other areas (Doling 2006; Krapp et al. 2020; Kucharska-Stasiak et al. 2022).

At the time of writing, there is a paucity of comprehensive understanding of the links between Europeanisation and housing studies. Europeanisation, a term used here

to explain and organize existing theories, concerns the various aspects of the incremental process of influencing or being influenced by different levels of government within the EU (or beyond) (Radaelli 2000, 2004; Radaelli and Exadaktylos 2008; Alpan 2021).

A review of the existing literature on multi-level European governance reveals a dearth of studies that address the topic of housing. In contrast, extant research tends to concentrate on either the national or local characteristics of housing systems (Allegra et al. 2020). The dearth of research in this field can be attributed to the fact that the EU lacks exclusive or shared competencies in the field of housing, and that the majority of housing provision policies are implemented at the local level. Nevertheless, numerous EU initiatives 'may exert an influence on housing policy and/or the housing system in member states' (Krapp et al. 2020, p. 144). These influences include policies to decarbonise the housing stock, EU legislation on state aid, liberalization, commodification, credit deregulation, and the EU strategy to combat poverty and social exclusion, among many others (Krapp et al. 2020; Tosics and Tulumello 2021). This phenomenon, which can be described as a 'spill-over effect', refers to the idea that these common influences could lead to increased pressure for common solutions in other highly political areas, even beyond the EU's sphere of influence, such as in housing (McCormick 2022).

The re-emergence of housing as a defining area of the EU social agenda has been proposed by several authors, including (Allegra et al. 2020; Delclós and Vidal 2021; Kucharska-Stasiak et al. 2022), in the wake of both the global financial crisis and the COVID-19 pandemic. In light of the aforementioned considerations, a comprehensive literature review was conducted with the objective of describing and better understanding the areas of the EU's influence in the field of housing. This entailed an examination of the diverse domains, approaches, mechanisms, and outcomes of Europeanisation (Exadaktylos and Radaelli 2012).

The domains of the influence of housing can be divided into three categories: politics (political actors, political parties, party system, and public opinion), policies (discourses, instruments, objectives, and practices), and polities (administrative structures; executive, legislative, and judicial authorities; democratic principles; and national systems of governance) (Radaelli 2000; Alpan 2021).

The approaches may be classified as top-down, bottom-up, or circular. The European Union (EU) is perceived in a variety of ways. Some view the EU as a fixed and teleological entity that the domestic level must adapt to, while others view it as a space for cross-border cooperation, interaction, learning, and networking among EU members. Furthermore, it can be perceived as a multifaceted governance system that encourages extensive cooperation and collaboration involving both public and private entities (Alpan 2021). With regard to the mechanisms that facilitate influence at the EU level, these include legislative, economic and fiscal integration, and cognitive integration through the exchange of best practices and European governance.

The preceding analysis enables the identification of gaps in the current body of knowledge and the formulation of a future research agenda. This should focus on the influences of member states, as well as the intersection between housing and other policy areas. Moreover, there is a dearth of research in the field of housing politics and polities that would permit an examination of the distribution and transfer of power in decision-making within the EU. It is essential to comprehend the strategic interplay between political actors (from disparate member states), EU institutions, and institutional rules (of the EU), and the consequences of such interactions (adoption, transformation, inertia, or even retrenchment).

This article is structured as follows: after this introduction, we will discuss and understand the research on Europeanisation in the field of housing. Next, taking into account the literature review, we will categorize the different approaches and mechanisms of Europeanisation. The EU's influences on housing will then be organized according to shifts in the European agenda. The conclusions section will summarize the answers to the research objectives and propose areas for future research.

## 2. Literature Review—Describing Europeanisation in the Field of Housing

*2.1. Research Methodology—Describing Europeanisation in the Field of Housing*

The research followed two interconnected paths, as demonstrated in Table 1.

**Table 1.** Literature review—research process.

| Research objectives | Describe and understand the Europeanisation in the field of housing. |
|---|---|
| Selection of filters | Social science research articles written in English<br>Keywords: "('European union' or EU or Europeani?ation) and housing and (policy or politics or polity)"<br>Scopus database, as it contains the majority of indexed journals and has a larger number of exclusive journals than Web of Science (Mongeon and Paul-Hus 2015; Pranckutè 2021) |
| Screening for inclusion | First, we have developed a frame of reference with six key areas: economic, environmental, legal, political, social and urban.<br>The title, abstract, and keywords of each document are then examined for inclusion. We only include research articles that gather relevant information on how the EU influences or is influenced by housing policies, politics, and polities at different levels of government.<br>Finally, we selected 40 documents for the next stage: the full-text review—the body of work was extracted and analysed in order to synthesize and report our findings. |

Source: own work.

In order to describe and understand Europeanisation in the housing field, specific filters were applied to identify relevant aspects. This approach aimed to ensure accuracy and avoid oversimplification, minimizing omissions and misunderstandings. In June 2023, a search was conducted for social science research articles written in English using the keywords 'European Union' or 'EU' or 'Europeanisation' and 'housing' and 'policy' or 'politics' or 'polity'. The Scopus database was selected as it includes the majority of journals indexed in the Web of Science and a greater number of exclusive journals (Mongeon and Paul-Hus 2015; Pranckutè 2021). A total of 383 documents were identified in the Scopus database and considered for the next stage of the process.

A screening process was conducted to determine which research articles to include, as shown in Table 2. Our frame of reference comprises six key areas: the economic, environmental, legal, political, social, and urban. The document's title, abstract, and the keywords of each document were screened for inclusion. Only research articles that gather relevant information on how the EU influences or is influenced by housing policies, politics, and polities at different levels of government were included. Following a comprehensive content review, 40 documents were selected for the next stage of the process, the full-text review. This body of work has been analysed and summarized in order to synthesize and report the findings.

A summary of the research methodology is provided below, based on the following sources (Petticrew and Roberts 2006; Siddaway et al. 2019; Xiao and Watson 2019):

**Table 2.** Areas, and respective authors and key themes.

| Authors | (Key) Themes | Key Areas |
|---|---|---|
| (Minnery 1996; Stephens 1999; Kasparova and White 2001; Norris and Winston 2011; Doling 2006; Aalbers 2009; Matos 2012; Boelhouwer 2014; Crespy 2016; Holleran 2017; Bohle 2018; Lestegás et al. 2018; Allegra et al. 2020; Tosics and Tulumello 2021) | Housing Market Dynamics;<br>European Economic Area, Economic and Monetary Union, Europeanisation of Mortgage Markets;<br>Fiscal Austerity Measures;<br>Liberalization, Commodification, Credit Deregulation;<br>Financialization of Housing. | Economic |

**Table 2.** *Cont.*

| Authors | (Key) Themes | Key Areas |
|---|---|---|
| (Winston 2007; Cerin et al. 2014; Delclós and Vidal 2021; Barbosa et al. 2022) | Carbon Neutrality; Decarbonization of Housing Stock; Energy-Efficient Housing Policies: Energy Performance Certificates, Zero-Energy Buildings. | Environmental |
| (Priemus 2006, 2008; Elsinga et al. 2008; Gruis and Priemus 2008; Magnusson and Turner 2008; Amann and Mundt 2010; Elsinga and Lind 2013; Boelhouwer 2014; Korthals Altes 2015; Mayoral and Pérez 2018) | Effect of EU Legislation on Social Rental Housing; EU Competition Policy; EU Consumer Law; EU Treaties; Single European Market Regulations. | Legal |
| (Chapman and Murie 1996; Doherty 2004; Priemus 2006, 2008; Czischke 2007; Allegra et al. 2020; Delclós and Vidal 2021; Tosics and Tulumello 2021) | European Governance—Best Management Practice; Principle of Sovereignty; Role of the State. | Political |
| (Chapman and Murie 1996; Giarchi 2002; Kleinman 2002; Kenna 2005; Mandic and Cirman 2012; Horváthová et al. 2016; Anderson et al. 2016) | Elderly Population; (Basic Human) Social Rights; Homelessness, Housing Disadvantaged People. | Social |
| (Vais 2009; Parysek 2010; Korthals Altes 2015; Purkarthofer 2019) | Regional/Urban Development; European Cohesion Policy; EU Urban Agenda—Soft Planning, Urban Regeneration. | Urban |

Source: own work.

## 2.2. Analysis of Selected Literature

The objective of this literature review is to describe and understand the manner in which the EU exerts influence over national housing policies, politics, and polities. In order to achieve this, a reference framework has been developed, comprising six key areas: economic, environmental, legal, political, and social (following 5), with the addition of urban (due to its housing-related agenda). These are the areas where the EU has exclusive or shared competence. The EU does not have competence over urban policy, but it shares competence over spatially relevant policies such as regional, environmental, or transport policies (Purkarthofer 2019). The analysis of the literature aimed to identify the approaches, mechanisms, and outcomes of Europeanisation, providing valuable insights for a more multi-level analysis focused on housing. Table 2 lists the authors and their main research themes.

### 2.2.1. Economic Influences: European Economic Area

First and foremost, the economic influences on housing are evident. These influences take the form of fiscal austerity, whereby budget deficits are limited to a certain percentage of GDP in order to adjust to global economic shocks and to ensure financial and macroeconomic stability (Allegra et al. 2020). The context of fiscal austerity is attributed to either the European single market and Economic and Monetary Union (EMU) or, later, the global financial crisis. (Stephens 1999; Kasparova and White 2001; Norris and Winston 2011), both of which are major areas of EU integration.

Minnery's research provides an example of the former (Minnery 1996). EU membership entails adherence to common budgetary targets. This represents one of numerous requirements that must be met. Consequently, the 'Swedish model' has been subjected to considerable pressure to adopt an even more market-oriented approach, which has had a 'spillover' effect on planning and housing policies.

The process of European integration has served as a catalyst for the marketisation of social services. This has been influenced by EU competition law and liberalization directives under the single European Economic Area (EEA) (Crespy 2016). The EEA and the European Monetary Union (EMU) are therefore two major influences (Minnery 1996; Stephens 1999; Norris and Winston 2011). This is accompanied by the Europeanisation of mortgage markets (Aalbers 2009; Norris and Winston 2011; Matos 2012; Boelhouwer 2014;

Bohle 2018). These common macroeconomic influences, namely financial liberalization, deregulation, and rescaling, could result in increased pressure for shared solutions in other highly political areas, even beyond the EU's sphere of influence, such as in housing.

In the context of the global financial crisis, austerity policies have significantly impacted the welfare state (Crespy 2016). EU countries have been influenced to reduce public spending and investment and have also increased the marketisation of financing and service provision in some policy areas. This has been the result of both voluntary choices and external pressures. As a consequence, the welfare state in many European countries has become (even) more residual, concentrating its efforts on a few target groups (Boelhouwer 2014). Since that time, and with greater impact today, housing's affordability has become a significant problem across the EU (and the world).

The global and EU frameworks for the free movement of capital and the provision of financial services, along with the availability of cheap credit and easier access to international finance for mortgage lenders, have led to the financialization of housing (Norris and Winston 2011; Bohle 2018; Lestegás et al. 2018). This refers to the understanding of housing as a market commodity. Holleran's findings indicate that the influx of foreign capital and EU loans has contributed to an exacerbation of over-investment, making it too easy to borrow for new housing (Holleran 2017). The authors concur on the central role of the housing sector in national and global economic equilibrium. The housing crisis has been identified as a significant contributing factor to the global financial and social crisis.

Nevertheless, the 'Swedish model' demonstrates that, even in the face of global economic influences and fiscal constraints, its system of local government remains relatively flexible, strong, and secure. This has led to the adaptation and innovation of EU policies to domestic circumstances (Minnery 1996).

A further illustration of the manner in which fiscal austerity in the EU is being addressed through adaptation and innovation is provided by (Stephens 1999). Stephens acknowledges that governments and housing providers are seeking alternative and complementary sources of funding, such as EU funding, in accordance with the principles of 'additionality' and 'subsidiarity'. The principle of 'additionality' pertains to EU funds, which are intended to supplement the public expenditure of member states, rather than replace it (Stephens 1999).

The (old) case studies in this literature review demonstrate that the European Commission has a strict policy against the use of EU funds for housing investments. In spite of this, there are some potential (limited) connections with member states' housing provision investments (Stephens 1999, pp. 726–28), including existing projects for non-housing purposes, but which are focusing on private housing development; existing housing organizations that have diversified and become eligible to receive EU funding; housing projects that do not have housing objectives, but where funds leak into housing; and housing-related projects that have leveraged additional funding.

The preceding examples demonstrate the manner in which the economic area of the EU exerts an influence on the housing provisions of its member states, while also ensuring adherence to the principles of subsidiarity and proportionality. The prioritization of market-oriented approaches may give rise to heightened tensions regarding the responsibilities of member states to provide housing, given that their role is limited to the correction and minimisation of market failures. The difficulties encountered by member states in addressing housing affordability independently, due to the constraints imposed by shared budgetary objectives and macroeconomic influences, highlight the need for a more integrated approach. It is of the utmost importance for member states to acknowledge the interconnectivity between housing's affordability and other policy domains, such as environmental, legal, or urban issues.

### 2.2.2. Environmental Influences: Carbon Neutrality

This section exemplifies another important influence. There is increasing evidence that the construction and renovation of buildings is being influenced by the EU. This

is driven by the EU's environmental agenda, as reflected in its legislative integration framework (Winston 2007; Cerin et al. 2014; Delclós and Vidal 2021; Barbosa et al. 2022). The authors concur that the energy renovation of the existing building stock is a critical factor in achieving Europe's decarbonization target. The EU's environmental directives, such as the Energy Performance of Buildings Directive from 2010 and the Energy Efficiency Directive from 2012, and funding, including the European Green Deal and the Recovery and Resilience Facility, are exerting significant pressure on Member States' housing policies and practices to decarbonise the building stock.

However, both EU funding and legislation have limitations and contradictions, particularly when focusing heavily on renovation, which presents challenges for housing affordability and inclusion (Delclós and Vidal 2021). Renovation may lead to increased housing prices (as indicated by Cerin et al. 2014), potentially worsening existing housing affordability issues.

Research indicates that the implementation of sustainable development policies related to housing varies across member states (Winston 2007; Cerin et al. 2014). Additionally, there is a lack of political commitment at both the national and local levels, with the absence of pressure from last-minute EU directives. This has resulted in the insufficient development of sustainable development policies related to housing (Winston 2007). The implementation of top-down legislative integration mechanisms may give rise to tensions between the housing provisions of member states and the EU's environmental agenda, particularly in relation to carbon neutrality.

### 2.2.3. Legal Influences: EU Legislation on (Social) Rental Systems

The legal framework that shapes member states' housing provision policies is influenced by both EU treaties and European Single Market rules. The aim of the EU is to create a single internal market without obstacles to the free movement of capital, goods, and services (in relation to the economic area) (Priemus 2006; Elsinga et al. 2008).

Member states must therefore ensure that their social housing policies align with the EU agenda, refrain from any distortion of competition, and make them 'Europe-proof' (Stephens 1999; Elsinga et al. 2008; Korthals Altes 2015). The European Commission generally advocates for the targeted allocation of social housing to those who are socially disadvantaged, a residual approach (Gruis and Priemus 2008).

This has the potential to have far-reaching consequences, underscoring the necessity for member states to possess the autonomy to shape their own policies (Priemus 2006). The Reform Treaty (2007) and EU competition policy have a significant impact on all EU housing associations. Consequently, it is incumbent upon each member state to delineate the parameters of its social housing providers and to draft regulations on legal state support in accordance with EU requirements, despite any political considerations to the contrary. Indeed, EU legislation has exerted an influence on the housing policies of its member states, even in the absence of a legal basis for a common approach to housing (Amann and Mundt 2010).

These influences are perhaps the most controversial in the literature, particularly with regard to the impact of EU legislation on (social) rental systems.

The tension between EU competition law (the European Transparency Directive, European competition law in conjunction with the EU Services Directive and the Altmark ruling) and national autonomy in the field of social housing has been explored by numerous authors, with particular focus on the Dutch case (Priemus 2006, 2008; Elsinga et al. 2008; Gruis and Priemus 2008; Korthals Altes 2015) or the Swedish case (Magnusson and Turner 2008), or even comparing both (Elsinga and Lind 2013). The Dutch and Swedish authorities have opted for a social rental sector that is open to a wide range of people and often in competition with private housing. This is contrary to the requirements of EU competition policy.

### 2.2.4. Political Influences: EU Institutions and European Networks

Most of the influences mentioned above are related to a top-down approach, such as the economic, legal, and environmental areas. The EU is presented as an independent variable to which member states must adapt (Alpan 2021). An exception to this is the case of judicial mobilization in Spain (Mayoral and Pérez 2018). Mayoral and Pérez argue that Spanish judges adopted an activist role by mobilizing and applying the CJEU jurisprudence to several cases to challenge national legislation against the backdrop of EU consumer law (Mayoral and Pérez 2018, p. 720). European consumer law creates a more protective situation for debtors facing bank repossessions than Spanish law. The authors conclude that EU consumer law is a deliberate strategy employed by national judges. This is evidenced by the fact that consumer protection law was the appropriate framework for assessing the imbalance between banks and mortgage debtors in legal proceedings. For Spanish judges, the impact of the EU came from cross-border cooperation, interaction, learning, and networking. In this case, Spanish judges were not passive, they have their own interests according to their socio-political context. This example is one of a few, and further analysis is necessary to gain a full understanding of the role and influences of member states (through their politicians and national systems of governance).

This leads to another lesson, which brings us to the political influences on the EU's role in housing. The role of the EU in housing has been, and remains, contradictory due to the existing 'asymmetry among politicians promoting market effectiveness and a policy promoting social welfare and equality' (Kucharska-Stasiak et al. 2022, p. 1). On the one hand, housing is regarded as a market commodity (as evidenced by the EU's economic and legal influences). The EU exerts state power and control towards the financialization of housing, which is a market-oriented approach. The role of the state is to correct market failures. On the other hand, housing is regarded as a basic social right, dependent on national decisions but influenced by the EU agenda (as evidenced by its social and urban influences).

As previously stated, although the EU does not possess exclusive or shared competence to develop housing policy, it does have responsibilities and develops programmes in policy areas that overlap with the operation and function of housing provision. According to Chapman and Murie, early ideas about the role of the EU focused on the following aspects (Chapman and Murie 1996):

- The European Commission plays a direct and formal role in developing and promoting dialogue between member states. This is carried out with the objective of facilitating the exchange of experience and good practice. In their examination of the long-term evolution of Portuguese housing policy, Allegra et al. demonstrate that the political and economic ideas, paradigms, and debates between the EU and national spaces have influenced Portuguese housing policy (Allegra et al. 2020).
- The pan-European networks that are involved in the fight against housing exclusion, such as CECODHAS or FEANTSA (Chapman and Murie 1996; Kleinman 2002), are worthy of mention. In this regard, Czischke's analysis of the network of European social housing providers (CECODHAS) indicates that the EU has gradually been invited to realize its need to establish a network-type governance structure to deal with the housing question (Czischke 2007). Nevertheless, there is a dearth of empirical studies that have examined the impact of network-type governance structures, including policy diffusion and learning (inherent in the informal meetings of housing ministers);
- The impact of the introduction of the Single European Act and European economic integration. Aspects of this have already been discussed. Doherty adds that, by the mid-1990s, it appeared that state power and control had receded, giving way to the market (Doherty 2004) This was largely due to the withdrawal of the state from direct involvement in housing provision in favour of home ownership schemes— an EU stealth housing policy (Doling 2006). In essence, he concludes, the role of the state is to act as a corrective to market failures. This is achieved through the

maintenance, regulation, and direction of the private housing market through interest rate manipulation, tax breaks, and other interventions and subsidies (Doherty 2004).

2.2.5. Social and Urban Influences: Social Rights and (Soft) Planning

A further lesson concerns the European Union's social concerns. As Kucharska-Stasiak et al. observed, the European Community was originally an economic and political project. However, due to the significant disparities that exist across the EU, attention has also been directed towards social factors. Even under EU economic and legal influences, the national context plays a crucial role in the provision of non-market housing for specific target groups, including the disadvantaged, the homeless and elderly, young people, and migrants (Mandic and Cirman 2012; Horváthová et al. 2016; Anderson et al. 2016). This is in order to prevent and tackle homelessness, which 'may reflect a market failure' (Anderson et al. 2016, p. 111).

It can be argued that the European debate, as predominantly described in the research, conceals divergent interpretations of the nature of housing markets and the scope of housing as a right or market commodity. Furthermore, these understandings transcend the conventional dichotomy between economic and social areas, which is often perceived as a dichotomy between European and national levels of analysis.

Housing policy choices vary between (and within) member states, but the EU strategy to combat poverty and social exclusion may have contributed to and influenced these policy choices (Czischke 2007), in line with the right to adequate housing as defined in the UN Declaration of Human Rights and the European Social Charter (Parysek 2010; Anderson et al. 2016).

This leads to the final lesson, which brings us to urban influences. The studies that have been carried out begin by highlighting that the EU does not have competence in the urban dimension. However, it does have shared competence over spatially relevant policies such as regional, environmental, and transport policies (Purkarthofer 2019). In accordance with the authors' findings, the aforementioned forces of EU membership (Vais 2009), EU structural funds with housing-related investments (Parysek 2010; Korthals Altes 2015), and the EU Urban Agenda are shaped and reinforced by the European Commission (Purkarthofer 2019). This influence is also exerted through the participation of member states in European governance initiatives, such as the Urban Agenda Partnership on Affordable Housing.

*2.3. Understanding Europeanisation in the Field of Housing*

The literature review provides a foundation for comprehending the multifaceted ways in which the European Union exerts influence on housing provision. It also elucidates the intricate interconnections between the various themes that shape this influence. Although there was some degree of overlap between the different areas of influence, they proved to be a useful tool in placing different influences into perspective. Furthermore, the existing literature demonstrates that, despite the EU's lack of competence in housing matters, EU legislation and policies may increasingly influence member states' housing provision policies. However, the existing literature on the European level and housing studies does not provide information on the EU's competence to support, coordinate, or complement the informal ministerial meetings of EU housing ministers or housing-related issues. Moreover, there is a dearth of data concerning the policy networks through which policy diffusion and learning can occur.

These influences have arisen from different shifts in the European agenda. They have been manifested in different approaches (top-down or bottom-up) and mechanisms (legislative, economic and fiscal, and cognitive integration), producing different housing outcomes as a result of opportunities (e.g., internal markets, EU structural funds). These may result in the accommodation, adaptation, or transformation of national housing priorities, and create tensions (through EU competition or environmental policies), leading to inertia or a retreat from the EU housing agenda. Yet, this remains to be studied in detail.

In light of the findings of the literature review, it is possible to categorize the various approaches and mechanisms of Europeanisation as follows.

The literature on Europeanisation distinguishes between top-down and bottom-up approaches. That is, between the EU 'as a fixed, categorical and teleological entity to which the domestic level must adapt' (Alpan 2021, p. 110) and the EU as a space for cross-border cooperation, interaction, learning, and networking among EU members. The vast majority of initiatives described in the literature review report a top-down approach. The exceptions are the groups and institutions of networks such as CECODHAS and FEANTSA (Giarchi 2002; Horváthová et al. 2016) and the example of the judicial mobilization case in Spain (Mayoral and Pérez 2018). Both are examples of bottom-up approaches. Supranational, national, and subnational actors (multi-level interactions) actively overlap, participate in the Europeanisation process, and empower and decentralize the accumulation of authority and power. The analysis of these approaches, namely bottom-up and circular approaches, could be valuable in introducing and discussing multi-scale and complex decision-making into housing studies.

In addition, the literature on Europeanisation and EU policies focuses on the mechanisms for increasing influence at the EU level (see Table 3):

- Legislative integration, also known as 'positive integration', is a process that aims to mitigate the adverse externalities that can result from market activities. These externalities may manifest in the domains of consumer protection, migration policy, or environmental and labour protection. In order to achieve this, a common European framework is developed that shapes a single European market.

**Table 3.** Mechanisms for enhancing influence at the EU level.

| | (Some) Tools of Government | Conflicts and Tensions between Principles, Rights, and Values | Housing Narrative |
|---|---|---|---|
| Legislative integration | Energy Performance of Buildings Directive (2010)<br>Energy of Efficiency Directive (2012)<br>European Green Deal (2019) | Principle of subsidiarity and proportionality, creating tensions between EU policies and national autonomy in housing<br>EU housing renovation targets conflating with universal values and rights of EU (human dignity, non-discrimination, and equality), leading to housing unaffordability and exclusion | Carbon neutrality energy-efficient housing policies: energy performance certificates, Zero-Energy Buildings |
| Economic and fiscal integration | EU Treaties (Treaty of Maastricht 1992), and Treaty on Functioning of the European Union (Treaty of Lisbon 2007)<br>Single European Act (1986) (European Economic Area, Economic and Monetary Union)<br>Recovery and Resilience Facility (2022) | Different conceptions of housing (as a public good, as a social right, or as a consumption asset) | Financialization of housing<br>Residual approach to social housing |
| Cognitive integration | Charter of Fundamental Rights of the European Union (2000)<br>EU Consumer Law (Directive 2008/48/EC on consumer credit agreements)<br>EU Urban Agenda (2016)<br>European Pillar of Social Rights (2017) | Different conceptions of housing (as a public good, as a social right, or as a consumption asset)<br>Homelessness 'may reflect a market failure' (Anderson et al. 2016, p. 111), conflating universal values and rights of EU: human dignity, equality, and solidarity. | The fight against poverty and social exclusion (homelessness, housing disadvantaged people)<br>Urban development and regeneration |

Source: adapted from (Pandžid 2021, p. 28) and based on (Korthals Altes 2015; Knill and Lechmkuhl 1999).

The approach is top-down in nature, prescribing direct institutional requirements that member states must comply with. The current underlying EU narrative in the field of housing is related to carbon neutrality, which is an environmental issue. This resulted in a conflict between EU policy and national autonomy in housing, in terms of the subsidiarity and proportionality principle. Moreover, the EU housing renovation targets may potentially conflict with the universal values and rights of the EU (human dignity, non-discrimination, and equality), which could result in unaffordability and exclusion.

- Economic and fiscal integration, or negative integration, is defined as the free movement of goods, services, labour, and capital. This integration is defined by the regulatory conditions that determine the extent of the competition for market access and operation. The principal mechanism of Europeanisation is the alteration of domestic opportunity resources and structures (Knill and Lechmkuhl 1999; Exadaktylos and Radaelli 2012).

As evidenced by the literature, the impact of EU legislation on social rental housing and the financialization of housing constitutes the EU housing narrative in economic and legal areas. In the absence of (shared) competencies at the EU level, Korthals Altes asserts that housing policy will continue to exhibit negative integration unless a single European housing market is created (Korthals Altes 2015, p. 341). Nevertheless, this prompts questions about the necessity of a European housing market and the principles agreed upon by EU member states.

- Cognitive integration, also known as framing integration, refers to changes in the beliefs and expectations of domestic actors (Radaelli 2000). This mechanism is the least researched, yet it is invaluable for comprehending the impact of the EU on the domain of housing. The literature identified three key themes of focus: the judgments of the European Court of Human Rights, the fight against poverty and social exclusion (homelessness, housing for disadvantaged people), and urban development and regeneration.

Furthermore, it is possible to organize the EU's influence on housing according to shifts in the European agenda.

- The phenomenon of a 'vertical shift' towards a supranational organization, as evidenced by the Treaties of the EU and the 'Functioning of the EU', as well as European competition and internal market rules. These measures have established a single set of rules within the EU area. For instance, when the responsibility for monetary policy was transferred to the European Central Bank, financial institutions were subject to a unified set of regulations and supervision, governments were constrained by debt and deficit limits, and barriers to the free movement of capital, labour, and services were removed. The top-down approach and economic and fiscal integration may be largely responsible for this change;
- Furthermore, the phenomenon of 'market shift' has emerged as a consequence of a top-down approach, economic and fiscal integration, and the establishment of the European Single Market and EMU. This shift has become more pronounced since the European debt crisis of 2009, leading to a more market-oriented approach. This has resulted in the financialization of housing, whereby housing is viewed as a market commodity. The state has assumed a more prominent role in the regulation of market failures, adopting a more residual approach to social housing;
- The phenomenon of a 'horizontal shift' (i.e., the influence of other EU policies on housing policy) is occurring in various areas of influence, including economic, environmental, legal, political and social, and urban areas.

Indeed, the EU exerts influence in a number of areas that are closely related to housing. It is therefore reasonable to conclude that housing will continue to be a key part of the European debate (Kleinman 2002). It can be posited that the influence of housing policy extends beyond the domain of housing policy itself, and that it also affects and impacts upon other policies (Doling 2006).

What remains is to test these influences, producing different housing outcomes, and to explore the missing links in the literature. There is a clear need for further research into the phenomenon of Europeanisation in the field of housing. The majority of the research in this field takes a normative view of EU (housing) policy, examining the influence of the EU on member states in isolation. While examining Europeanisation as an independent variable in isolation may not fully account for all domestic changes within the EU, it is important to recognise that member states are not passive actors in the face of EU demands for change. Rather, they have their own interests, shaped by their socio-political contexts, which may lead to support or resistance to EU-induced reforms. In other words, there is a need for a more analytical understanding of the ways in which member states influence and are influenced by housing-related policies, as well as the intersections between housing policy and other policy areas. It would be beneficial to consider and discuss political perspectives, such as the distribution and transfer of power in decision-making, as well as the political and social mechanisms of democratic interactions.

## 3. Conclusions

The European Union (EU) exerts a profound, yet often overlooked, influence on housing provision across its member states. The objective of our research was to gain a deeper insight into the influence of the EU on housing provision at different levels of government.

A review of the relevant literature reveals several key areas where the EU exerts influence. These include economic and fiscal policy, environmental regulations, legal frameworks, political networks, social policies, and urban development and regeneration strategies (see Tables 2 and 3).

Each of these areas is characterized by a distinct set of themes that influence the manner in which housing is provided and managed within member states. The influences on housing provision in member states can be divided into the following:

- Those resulting from the definition of the economic, fiscal, and institutional framework of housing market contracts (e.g., economic and legal areas);
- Those resulting from policy decisions regarding housing tenures, including limitations to state aid such as subsidies, grants, and tax breaks, and macroeconomic influences such as financial liberalization and the deregulation of the banking/mortgage sector, which favours home ownership schemes;
- Those resulting from housing justice frameworks, such as homelessness and vulnerable people.

Examples of these spill-over effects include policies to decarbonise the housing stock, EU legislation on state aid, liberalization, commodification, credit deregulation, and the EU strategy to combat poverty and social exclusion, among many others (Krapp et al. 2020; Tosics and Tulumello 2021). These policies establish the parameters within which housing provision is conducted at lower levels.

The influences in question originate from different stages of the EU's integration. These include the 'negative' integration under treaties (of the EU and later, the Functioning of the EU) and European Single Market regulations. These translate into the conflicts and tensions between the EU and national autonomy in housing policy, which can be described as a (re)scaling from national sovereignty to an EU multi-level approach.

Furthermore, there are those related to the overlapping influences of 'positive' and 'negative' integration, namely the 'Green Deal' (2019) and the 'Recovery and Resilience Facility' (2022). This second overlap translates into the conflicts and tensions between EU housing renovation targets and housing affordability and inclusion, which can be described as an (in)equity dilemma.

In both cases, there is a discrepancy in opinion regarding the nature of housing. This can be observed in the differing views on whether housing should be considered a public good, a social right, or a market commodity. The divergence is not limited to the housing provision choices made by individual member states, but also extends to

the differing approaches taken by EU policy instruments. The current financialization of housing—where housing is seen as a market commodity—and the limited role of the state in correcting or mitigating market failures may exacerbate the housing crisis and potentially undermine the EU's goal of a competitive but social market economy (aiming at social justice and progress) (Geiger et al. 2015).

Another aspect relates to the mechanisms for enhancing influence at the EU level. These include legislative integration (or 'positive integration', which aims to limit the negative externalities arising from market activities), particularly in relation to environmental and legal areas; economic and fiscal integration (or 'negative integration', which aims at the free movement of goods, services, labour, and capital), in relation to economic and legal areas; and cognitive integration (or 'framing integration', which refers to changes in the beliefs and expectations of domestic actors), mainly in relation to political, social, and urban areas.

Although the majority of the literature on this focuses on the influence of the EU on its member states, it is important to recognise that the provision of housing is not a one-way process. Furthermore, the decisions made by member states can also impact the EU agenda. Consequently, there is a need for a more nuanced understanding of these dynamics.

Further analysis is warranted. Indeed, it could be argued that housing policy is not only affected but also influences and impacts other policies (Doling 2006). Moreover, most research considers the EU 'as a fixed categorical and teleological entity to which the domestic level must adapt' (Alpan 2021, p. 110). The EU, as an independent variable, cannot explain all the domestic changes in EU member states. Member states cannot be seen as passive to EU demands for change. In other words, there is a need for a more analytical understanding of the ways in which member states influence housing politics and polity. There is also a need for a more analytical understanding of where housing intersects with—and influences—other policy areas. How do member states' decisions about housing affect the EU agenda? What factors (governance systems, NGOs, political actors, party systems, public opinion) are decisive and with what consequences (adaptation, transformation, inertia, or retrenchment)? These questions are pertinent and fall within the scope of a more in-depth analysis.

**Author Contributions:** Conceptualization, methodology, writing—original draft preparation, writing—review and editing: J.A. and P.C.; Supervision: P.C. All authors have read and agreed to the published version of the manuscript.

**Funding:** This research received no external funding.

**Institutional Review Board Statement:** Not applicable.

**Informed Consent Statement:** Not applicable.

**Data Availability Statement:** No new data were created or analyzed in this study. Data sharing is not applicable to this article.

**Conflicts of Interest:** The authors declare no conflicts of interest.

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
