# Peer review of "Europeanisation in the Field of Housing: Its Areas of Influence, Different Approaches, Mechanisms, and Missing Links"

_socsci, doi:10.3390/socsci13050268_

Round 1

Reviewer 1 Report

Comments and Suggestions for Authors

This is an interesting survey of the impact of European integration on housing regimes/policy, and potentially, the influence of housing regimes/policy on the European Union.

The authors stress the limits of direct influence because the EU has no mandate to adopt legislation in this field. However, the authors point out that the Green Deal relies on de carbonizing buildings, including residential dwellings and that many EU member states face a housing affordability crisis.

One area that is not developed in the paper is the existence of policy networks of national housing experts who meet regularly, trade information, and form, to some extent, an epistemic community. These networks (there could be more than one) are especially active in the field of social housing/affordable housing and receive funding from the Commission to publish joint reports, as well as subsidies to experiment with new forms of design and construction that is accessible to low-income households and incorporates the newest green/energy efficiency technology. 

In short, there may also be a process of policy diffusion/learning taking place, mostly because many member states cope with similar social issues related to rising rents, rising housing prices, and shrinking public housing sector. 

This is a great survey, though the paper may benefit from examining the way in which  European policy networks and advocacy groups contribute to a   process of "Europeanization." 

Author Response

We would like to express my gratitude for your review of this manuscript. Please find the responses and corresponding revisions highlighted in the resubmitted file.

Comments:

[2a] One area that is not developed in the paper is the existence of policy networks of national housing experts who meet regularly, trade information, and form, to some extent, an epistemic community. These networks (there could be more than one) are especially active in the field of social housing/affordable housing and receive funding from the Commission to publish joint reports, as well as subsidies to experiment with new forms of design and construction that is accessible to low-income households and incorporates the newest green/energy efficiency technology.

[2b] In short, there may also be a process of policy diffusion/learning taking place, mostly because many member states cope with similar social issues related to rising rents, rising housing prices, and shrinking public housing sector.

[2c] This is a great survey, though the paper may benefit from examining the way in which European policy networks and advocacy groups contribute to a process of "Europeanization.

Response

Thank you for bringing this matter to our attention. We acknowledge and agree with this comment. Therefore, we have included a third contribution related to future research agendas. This contribution should focus on understanding the impact of member states and the intersection between housing and other policy areas. Furthermore, we add that “there is a lack of research in the field of housing politics and polities that would enable us to examine the power distribution and transfer in decision-making within the EU. This involves understanding the strategic interaction between political actors (from different member states), EU institutions and institutional rules (of the EU), and the consequences of such interaction (adoption, transformation, inertia or even retrenchment)” (p. 2, l. 92-97).

More examples: 

“However, there are few empirical studies that have examined the impact of cognitive Europeanization on housing policy, including policy diffusion and learning, within the policy networks of national housing providers (e.g. inherent in informal meetings of housing ministers).” (p. 7, l. 287-290); 

“Little research has been done on the impact of informal ministerial meetings, such as those held by the EU Ministers Responsible for Urban Matters, which resulted in the EU Agenda.” (p. 8, l. 312-314).

“However, the literature on the European level and housing studies does not provide information on the EU's competence to support, coordinate or complement informal ministerial meetings of EU housing ministers or housing-related issues. In addition, there is insufficient information on the policy networks through which policy diffusion and learning can take place, and this may need to be reconsidered.” (p. 8, l. 333-338)

Reviewer 2 Report

Comments and Suggestions for Authors

The manuscript titled "The Europeanization in the Field of Housing: Areas of Influence, Different Approaches, Mechanisms, and Current Scenario" offers an useful exploration into the indirect yet significant influence of European Union (EU) policies on housing within member states. The content analysis of European Commission documents is particularly relevant for its methodological consistency and for supporting the thesis that housing has re-emerged as a critical area of interest within EU policy circles. This paper, therefore, stands as a good contribution to both Europeanization studies and housing policy literature. I have a few comments that I hope will be to strengthen it for future publication. By addressing these points, I believe the paper could significantly improve in coherence, clarity, and overall impact, making a substantial contribution to the literature on Europeanization and housing policy.

Key Points:

·       The coverage of the paper is very broad, touching on several crucial aspects of Europeanization in the housing sector. However, I believe the scope might be overly ambitious for a single manuscript. This breadth has led to a certain disjointedness between the literature review and the document analysis sections. In my view, the paper could be more effective and coherent if split into two separate papers. One could focus on the literature review to delve deeper into existing research and theoretical frameworks, while the other could concentrate on a detailed analysis of the selected EU documents. Such a division would allow each paper to offer a more focused and thorough exploration of its subject matter.

·       In terms of clarity, I've noticed some terms, specifically 'additionality' and 'subsidiarity,' that are crucial to understanding the EU's fiscal and policy-making processes, lack clear definitions. Clarifying these terms would significantly aid readers unfamiliar with EU jargon, which would be good to improve the paper's accessibility.

·       A critical examination of one of the statements in the paper reveals a potential oversight. The assertion that "there has been little research on how urban areas influence the housing domains" seems to contradict the wealth of studies published in journals such as Urban Studies, Housing Studies, and European Planning Studies, to cite a few. A more nuanced or revised statement could better reflect the existing research landscape and its implications for the paper's arguments.

·       A small paper restructuring would provide readers with a clearer understanding of the research context and objectives from the outset. I find that the paper could benefit from better signposting, especially within the literature review section. This could improve the paper's readability, making it easier for readers to follow the arguments and understand the connection between different sections. Additionally, the discussion about the literature gap, currently located on page 9, might be more appropriately placed in the introduction.

Minor point:

·       I recommend a revision of the numbering in some sections to ensure consistency throughout the paper. This change might seem small but can significantly enhance the professional presentation and readability of the manuscript.

Comments on the Quality of English Language

N/A

Author Response

I would like to express my gratitude for your review of this manuscript. Please find the responses and corresponding revisions highlighted in the resubmitted file.

[2] The coverage of the paper is very broad, touching on several crucial aspects of Europeanization in the housing sector. However, I believe the scope might be overly ambitious for a single manuscript. This breadth has led to a certain disjointedness between the literature review and the document analysis sections. In my view, the paper could be more effective and coherent if split into two separate papers. One could focus on the literature review to delve deeper into existing research and theoretical frameworks, while the other could concentrate on a detailed analysis of the selected EU documents. Such a division would allow each paper to offer a more focused and thorough exploration of its subject matter.

Response: 

Thank you for bringing this matter to our attention. We acknowledge and agree with this comment. Accordingly, we have modified the scope of the research paper to "delve deeper into existing research and theoretical frameworks". Subsequent research can focus both on a detailed analysis of the selected EU documents and on moving towards a research agenda of theory-informed, actor-related analysis of housing politics/polities. This could be done through both semi-structured interviews and surveys with local (housing) authorities in order to identify the logics, patterns and socio-political mechanisms in a specific context (the Portuguese one, under the 'Recovery and Resilience Facility').

[3] · In terms of clarity, I've noticed some terms, specifically 'additionality' and 'subsidiarity,' that are crucial to understanding the EU's fiscal and policy-making processes, lack clear definitions. Clarifying these terms would significantly aid readers unfamiliar with EU jargon, which would be good to improve the paper's accessibility.

Response: 

We agree. We have therefore added these (and other) definitions to improve the readability of the paper.

“(…) under the principle of subsidiarity and proportionality. Both principles define the relationship between the EU and the Member States. The first principle states that the EU should only undertake tasks that are necessary or that it can perform most effectively. The second principle states that the EU should not exceed what is required to achieve its objectives [8,9].” (p. 1, l. 38-42)

“The principle of 'additionality' refers to EU Funds, which are meant to supplement the public expenditure of member states, rather than replace it [24].” (p. 5, l. 184-186)

[4] A critical examination of one of the statements in the paper reveals a potential oversight. The assertion that "there has been little research on how urban areas influence the housing domains" seems to contradict the wealth of studies published in journals such as Urban Studies, Housing Studies, and European Planning Studies, to cite a few. A more nuanced or revised statement could better reflect the existing research landscape and its implications for the paper's arguments.

Response: 

We are in agreement with this comment. This is consistent with the idea of “there are few empirical studies that have examined the impact of cognitive Europeanization on housing policy, including policy diffusion and learning, within the policy networks of national housing providers (e.g. inherent in informal meetings of housing ministers).” (p. 7, l. 287-290). We have therefore revised the statement to better reflect the missing links in the research and their impact on the paper's arguments “little research has been done on the impact of informal ministerial meetings, such as those held by the EU Ministers Responsible for Urban Matters, which resulted in the EU Agenda” (p. 8, l. 312-314).

5a] · A small paper restructuring would provide readers with a clearer understanding of the research context and objectives from the outset. I find that the paper could benefit from better signposting, especially within the literature review section. This could improve the paper's readability, making it easier for readers to follow the arguments and understand the connection between different sections. 

Response: 

It is acknowledged that the literature review section requires guidance for the reader. Therefore, certain words have been highlighted to aid comprehension.

[5b] Additionally, the discussion about the literature gap, currently located on page 9, might be more appropriately placed in the introduction

Response: 

The introduction section has identified a gap in the literature, as opportunely suggested. 

“The third contribution pertains to the future research agenda, which should concentrate on comprehending the influences of member states as well as the intersection between housing and other policy areas. Furthermore, there is a lack of research in the field of housing politics and polities that would enable us to examine the power distribution and transfer in decision-making within the EU. This involves understanding the strategic interaction between political actors (from different member states), EU institutions and institutional rules (of the EU), and the consequences of such interaction (adoption, transformation, inertia or even retrenchment).” (p. 2, l. 90-97)

Reviewer 3 Report

Comments and Suggestions for Authors

- I suggest to move the description of the literature review phases (pag. 3) to a separate section as part of the methodology, assuming that the authors have performed a systematic literature review.  

- Remove or rename section 2.2. I don't think the word summary should appear there. Section 2.2. should be incorporated to 2.1. The theoretical framework should embed both influences and mechanisms and critically discuss how they interact. So far, they are detached from each other.

- Section 3. It is not clear what the actual meaning of "current scenario behind the inclusion of housing in EU policies" is. Isn't this section presenting the method? 

- Generally, the paper presents an extensive collection of information but is somehow pedantic. I encourage the authors to put some more critical reflections in the paper and let the argument emerge more explicitly and with authors' own words.

- I found myself lost while reading this paper since it is very dense and difficult to follow. Simplify the sentences and removing the unnecessary information would help a lot.    

Comments on the Quality of English Language

English language is understandable and generally correct.    

Author Response

I would like to express my gratitude for your review of this manuscript. Please find the responses and corresponding revisions highlighted in the resubmitted file.

[1] I suggest to move the description of the literature review phases (pag. 3) to a separate section as part of the methodology, assuming that the authors have performed a systematic literature review.

Response:

To enhance the paper's comprehensibility, it is recognized that a dedicated methodology section is essential to offer guidance to the reader. The methodology is described in the following manner.

“2.1. Research Methodology - Evaluating the Europeanization in the Field of Housing

The research followed two interconnected paths, as demonstrated in Table 1. 

To describe and understand Europeanization in the housing field, we applied specific filters to identify relevant aspects. This approach aimed to ensure accuracy and avoid oversimplification, minimizing omissions and misunderstandings. In June 2023, a search was conducted for social science research articles written in English using the keywords 'European Union' or 'EU' or 'Europeani?ation' and 'housing' and 'policy' or 'politics' or 'polity'. The Scopus database was chosen as it includes most journals indexed in the Web of Science and has a greater number of exclusive journals [20,21]. A total of 383 documents were identified in the Scopus database and considered for the next step.

A screening process was conducted to determine which research articles to include, as shown in Table 2. The frame of reference comprises six key areas: economic, environmental, legal, political, social, and urban. The document's title, abstract, and keywords of each document were screened for inclusion. Only research articles that gather relevant information on how the EU influences or is influenced by housing policies, politics, and polities at different levels of government were included. After conducting a comprehensive content review, we have selected 40 documents for the next stage: the full-text review. We analyzed and summarized the body of work to synthesize and report our findings.” (p. 3, l. 105-123)

[2]- Remove or rename section 2.2. I don't think the word summary should appear there. Section 2.2. should be incorporated to 2.1. The theoretical framework should embed both influences and mechanisms and critically discuss how they interact. So far, they are detached from each other.

[3]- Section 3. It is not clear what the actual meaning of "current scenario behind the inclusion of housing in EU policies" is. Isn't this section presenting the method?

Response: 

Thank you for bringing these matters to our attention. We agree with these comments and have modified the scope of the research paper to provide a more critical analysis of existing research and theoretical frameworks. Subsequent research can focus both on a detailed analysis of the selected EU documents and on moving towards a research agenda of theory-informed, actor-related analysis of housing politics/polities. This could be done through both semi-structured interviews and surveys with local (housing) authorities in order to identify the logics, patterns and socio-political mechanisms in a specific context (the Portuguese one, under the 'Recovery and Resilience Facility').

[4] - Generally, the paper presents an extensive collection of information but is somehow pedantic. I encourage the authors to put some more critical reflections in the paper and let the argument emerge more explicitly and with authors' own words.

[5] - I found myself lost while reading this paper since it is very dense and difficult to follow. Simplify the sentences and removing the unnecessary information would help a lot.

Response: 

We concur with this observation and have therefore revised the text in order to provide a more critical and clear narrative.

Round 2

Reviewer 3 Report

Comments and Suggestions for Authors

1) From p. 4 to p.8: please discuss each category of influence (economic, environmental etc.) as a separate subsection, instead of underlying the text.

2) Please go through my previous comment again, as it has not been fully incorporated in the new version. Enhance the narrative and be more critical.

Generally, the paper presents an extensive collection of information but is somehow pedantic. I encourage the authors to put some more critical reflections in the paper and let the argument emerge more explicitly and with authors' own words.

I found myself lost while reading this paper since it is very dense and difficult to follow. Simplify the sentences and removing the unnecessary information would help a lot.

Comments on the Quality of English Language

Moderate editing of English language required

Author Response

[1] From p. 4 to p.8: please discuss each category of influence (economic, environmental etc.) as a separate subsection, instead of underlying the text.

Response: 

In order to enhance the intelligibility of the document, it is acknowledged that each category of influence should be allocated to a distinct section. 

[2] Please go through my previous comment again, as it has not been fully incorporated in the new version. Enhance the narrative and be more critical.

Generally, the paper presents an extensive collection of information but is somehow pedantic. I encourage the authors to put some more critical reflections in the paper and let the argument emerge more explicitly and with authors' own words.

I found myself lost while reading this paper since it is very dense and difficult to follow. Simplify the sentences and removing the unnecessary information would help a lot.

Response: 

We concur with this observation and have, therefore, meticulously revised the text to provide a more critical and lucid narrative, as detailed below.

  • The paragraphs below provide examples of text that have undergone a process of revision. The writing of these paragraphs has been streamlined to enhance readability.

Introduction, p. 2, lines 64-93: paragraphs that serve to elucidate the rationale behind the narrative of this article, considering the following points:

  • The housing crisis represents a central concern in the contemporary era; 
  • The EU does not possess exclusive or shared competence in this area;
  • The EU exerts influence through the implementation of policies in other areas; 
  • There is a paucity of comprehensive understanding of the links between Europeanisation and housing studies
  • The future research agenda should prioritize the examination of the influence of Member States and their intersections with other policy areas. Moreover, the politics and polities of housing at the EU level should be considered.

p. 6, lines 217-229, paragraphs which provide a summary of the environmental influences.

p. 6, lines 231-257, the argument put forth is that Member States must ensure that their social housing policies align with the EU agenda, refrain from any distortion of competition, and make them 'Europe-proof'.

Conclusions, pp. 11-12, lines 451-519, considering the following points: 

  • The EU exerts a multifaceted influence on housing, encompassing economic and fiscal policy, environmental regulations, legal frameworks, political networks, social policies, and urban development strategies;
  • The EU's influence on housing provision arises from different stages of EU integration, including the "negative" integration, and the overlapping influences of the EU's policies;
  • These different stages of integration reflect the conflicts and tensions between EU and national autonomy in housing policy;
  • A more nuanced understanding of these dynamics is required, including an analysis of how housing policy influences other policies, how Member States influence housing provision policy, and where housing intersects with and influences other policy areas.

Questions for Further Analysis

- How do Member States' decisions on housing affect the EU agenda? 

- What factors (governance systems, NGOs, political actors, party system, public opinion) are decisive and with what consequences (adaptation, transformation, inertia or retrenchment)?

  • Furthermore, a critical analysis of the existing literature was conducted, as presented below.

p. 5, lines 195-205

“The preceding examples demonstrate the manner in which the economic area of the EU exerts an influence on the housing provisions of its Member States, while also ensuring adherence to the principles of subsidiarity and proportionality. The prioritization of market-oriented approaches may give rise to heightened tensions regarding the responsibilities of Member States in providing housing, given that their role is limited to the correction and minimisation of market failures. The difficulties encountered by Member States in addressing housing affordability independently, due to the constraints imposed by shared budgetary objectives and macroeconomic influences, highlight the need for a more integrated approach. It is of the utmost importance for Member States to acknowledge the interconnectivity between housing affordability and other policy domains, such as environmental, legal, or urban issues.”

p. 8, lines 231-325,

“It can be argued that the European debate, as predominantly described in the research, conceals divergent interpretations of the nature of housing markets and the scope of housing as a right or market commodity. Furthermore, these understandings transcend the conventional dichotomy between economic and social areas, which is often perceived as a dichotomy between European and national levels of analysis.”